# Toward Prevention of Doping in Youth Sport: Cross-Sectional Analysis of Correlates of Doping Tendency in Swimming

**DOI:** 10.3390/ijerph16234851

**Published:** 2019-12-02

**Authors:** Dorica Sajber, Dora Maric, Jelena Rodek, Damir Sekulic, Silvester Liposek

**Affiliations:** 1Faculty of Sport, University of Ljubljana, 1000 Ljubljana, Slovenia; 2PhD Program in Health Promotion and Cognitive Sciences, Sport and Exercise Research Unit, Department of Psychological, Pedagogical and Education Sciences, University of Palermo, 90144 Palermo, Italy; 3Faculty of Kinesiology, University of Split, 21000 Split, Croatia; 4University of Maribor, 2000 Maribor, Slovenia

**Keywords:** performance enhancement, puberty, achievement, knowledge

## Abstract

Doping is recognized as one of the most important problems in sports, but a limited number of studies have investigated doping problems in youth athletes. This study aimed to evaluate doping tendency (potential doping behavior (PDB)) and correlates of PDB in youth age swimmers. The participants were 241 competitive swimmers (131 females; 15.3 ± 1.1 years of age, all under 18 years old). Variables included predictors and PDB (criterion). Predictors consisted of sociodemographic factors (gender and age), sport-related variables (i.e., experience in swimming and sport achievement), variables explaining coaching strategy and training methodology, consumption of dietary supplements (DS), knowledge about doping, and knowledge about sports nutrition and DS (KSN). In addition to the descriptive statistics and differences between genders, a multinomial regression using PDB as the criterion (negative-, neutral-, or positive-PDB, with a negative-PDB as the reference value) was calculated to define associations between predictors and criterion. With only 71% of swimmers who declared negative-PDB results indicated an alarming figure. Boys with better KSN were more negatively oriented toward positive-PDB (OR: 0.77, 95%CI: 0.60–0.95). In girls, lower competitive achievement was evidenced as a risk factor for neutral-PDB (OR: 0.39, 95%CI: 0.24–0.63). Also, higher neutral-PDB (OR: 0.88, 95%CI: 0.81–0.96) and positive-PDB (OR: 0.90, 95%CI: 0.83–0.99) were identified in girls who began with intensive training in younger age. Because of the alarming figures of PDB, there is an evident need for the development of systematic antidoping educational programs in youth swimming. In doing so, focus should be placed on girls who began intensive training at an earlier age and those who did not achieve high competitive results.

## 1. Introduction

Doping behavior is considered as one of the most important problems in sport, not only because of the detrimental health consequences of consumption of doping substances, but also because doping corrupts the essence, image, and value of sport [1,2]. Although the World Antidoping Agency (WADA) has been exploring and developing the most efficient ways to improve doping statistics, data from WADA antidoping rule violation reports do not provide evidence of decrease (2015: 1.26%; 2016: 1.60%; 2017: 1.43%; positive analytical findings), and WADA approximates that a true number exceeds 10% [3,4]. Furthermore, doping extends beyond regulated elite sports into lower levels and even outside of the sporting context, where it may be used to enhance the development of a person’s physique; however, the concerning finding is that doping affects younger age groups [5]. Doping use among adolescent athletes is concerning, as it is known that doping exploitation at this crucial age leads to a longer period of substance use and, consequentially, to a higher potential for severe negative health consequences [6]. In addition, adolescence is a period of both biological and social transition, with changes in cognitive, moral, social, and emotional development. This period of life involves many changes and has long been portrayed as a time of stress and hardship, but also provides a unique opportunity for inculcating healthy attitudes and behaviors, which makes it a fruitful period for introducing antidoping strategies [7,8,9]. Adolescence is considered as critical period for preventing substance misuse [10], and not surprisingly is recognized as critical period for prevention of doping behavior in sport as well [11,12,13].

Specifically, British and Italian studies have indicated that young athletes who are convinced of the necessity of supplementation for sporting success are also more likely to express permissive attitudes [12,13]. In studies reporting the prevalence of anabolic steroid use, Faigenbaum et al. suggested that a small but significant portion of middle school females and males have used anabolic steroids [14], and Corbin et al. and Dodge et al. established that steroids are more readily available to males, who also reported knowing more steroid users compared to females [15,16]. Chan et al. analyzed data collected from a sample of young elite and sub-elite Australian athletes, and came to the conclusion that athletes who were autonomously motivated toward sports were more likely to be autonomously motivated toward doping avoidance, which has been shown to be the most favorable motivational orientation for the formation of a positive attitude, perceived behavioral control, subjective norms, and the intention of doping avoidance [17]. In studies examining social-cognitive mechanisms, Lucidi et al., and Zelli et al. implied that doping intentions in adolescents increased with stronger attitudes about doping, a stronger conviction that doping use can be justified, stronger beliefs that significant others would approve of the use, and a lowered capacity to resist situational pressure or personal desires, as well as that adolescent doping use is regulated by a doping-specific belief system [8,18].

In general, many studies that have been conducted regarding either actual or potential doping behavior (PDB) in sports have focused on identifying factors associated with doping behavior [19,20,21]. Although there are numerous personal and environmental factors that influence doping behavior, the unique relationship that exists between coaches and athletes is one of the factors that should be observed as an important determinant of athletes’ attitudes toward doping. This is mostly because of the coaches’ influence on athletes’ behaviors and actions, making coaches potential agents in the prevention of doping [19,22]. One promising approach is the identification of associations that exist between different types of coaching strategies and training methodologies (CS&TM) and PDB in athletes. In brief, CS&TM describes sets of training methods and strategies that coaches use throughout the sports training process to improve the athletes’ physiological capacities and sport-specific skills [23]. Indeed, a recent study identified (i) athletes who recognized their training as monotonous and oriented toward volume and (ii) athletes who perceived their coaches as being indifferent to the evaluation of their training goals and achievement as being simultaneously more prone to doping [23]. Moreover, variables of CS&TM were stronger correlates of PDB than any other evaluated predictor (e.g., sociodemographics and sport factors), which indicates the potential applicability of studying aspects of CS&TM in order to prevent doping behavior in sports. However, this issue was not studied in youth athletes.

Swimming is considered to be one of three elementary (basic) sports (together with athletics and artistic gymnastics), which directly contributes to its global popularity. Sports results (i.e., achievement) in swimming are determined by the interaction of morphological, physiological, psychological, and technical factors based on individual genetic endowments, which directly defines the high physiological and psychological stress placed on athletes during a relatively long career [24,25,26]. At the same time, great popularity and high demands are probable reasons why in 2016 WADA report placed swimming among 8% of the most doping contaminated sports [3]. Therefore, an antidoping policy in swimming is a high priority, which is directly recognized by the official authorities in swimming, including the international swimming federation, FINA [27]. As a result, recent studies have followed FINA position statements and guidelines and, consequently, studied factors that influence doping behavior in swimming [23,28,29]. However, no study has examined factors associated with doping behavior specifically in youth swimmers. This fact is important, as adolescence is a critical time when young people are formulating attitudes and laying the groundwork for their future health and well-being [7,8,30]. That being said, it is clear that adolescence is an ideal period for inculcating healthy attitudes and behaviors and for conducting antidoping strategies.

This study aimed to provide insight into potential doping behavior (PDB) in youth swimming, as well as associations that may exist between certain correlates (i.e., sociodemographic factors, sport-related variables, factors of CS&TM, consumption of dietary supplements, and knowledge about doping and sports nutrition (predictors)) and PDB (criterion), specifically in youth-aged swimmers (under 18 years old). We hypothesized that studied predictors will be significantly associated with PDB in youth swimmers.

## 2. Materials and Methods

### 2.1. Study Design

In this study, we used cross-sectional design to identify (i) gender differences in studied variables including PDB and (ii) associations between predictors and criterion—potential doping behavior. All participants were tested at the same occasion in a time frame of 2 days by previously validated questionnaires examining predictors and criteria. The respondents were tested in groups of at least five athletes, and testing was conducted in the local language. All participants were informed that the survey was anonymous, they could refuse to participate, and they could leave some of the questions or the entire questionnaire unanswered. Because all of the participants were younger than 18 years of age, written parental consent was obtained for all the participants in the study. Also, participants were informed that returning the completed questionnaire was considered consent to participate in the study. After testing, the questionnaires were placed in a closed box, which was opened the day after the testing. The response rate was high: only 1 athlete returned an unanswered questionnaire.

### 2.2. Participants

Elite junior age swimmers from Slovenia were tested during their National Championship held in Maribor, Slovenia 2017. The invitation to participate in the study was sent by the national swimming federation and was accepted by all competitors [23,28]. The study was originally approved and initiated by the national swimming federation and was approved by the Ethical Board of the University of Split, Faculty of Kinesiology (EBO: 2181-205-02-05-14-004). The final sample comprised 242 swimmers (131 females, mean 14.4 ± 1.2 years; 111 males, mean 15.3 ± 1.1 years) who were all younger than 18 years at the moment of testing. The sample included all youth age swimmers (<18 years) who participated in the National Championship for the year of 2017. Therefore, we may say that the total population of competitive youth age swimmers from the country was included/represented.

### 2.3. Variables and Testing

The previously validated questionnaires were used to collect data about the predictors and the criterion (PDB) [23,31,32]. Based on previous investigations in the field, the predictors included sociodemographic data, sport factors, consumption of dietary supplements, knowledge about sports nutrition (KSN), knowledge about doping (KD), doping-related factors, and variables explaining CS&TM [23,31,32,33].

The sociodemographic data included age (in years) and gender. Sport factors were assessed by using questions about (i) the athlete’s experience in swimming (in years), (ii) the age when the athlete began training two times a day, (iii) the age when the athlete started training 8 or more times a week, and (iv) competitive results achieved in (iv-a) non-Olympic events (25-m pool) and (iv-b) Olympic events (50-m pool) (i.e., “Regional level medalist”, “National Championship—finalist”, “National Championship—medalist”, “International competition—finalist”, and “International competition—medalist”). Doping-related factors were evaluated by asking participants their opinions about (i) the occurrence of doping in swimming (“I don’t think doping is used in swimming”, “Not sure about it”, “Occurs, but rarely”, “Doping occurs often”), (ii) the number of times they have been tested for doping (“Never tested for doping”, “Once or twice”, “Three times or more”, “>5 times”), (iii) their potential doping behavior (“I would engage in doping if it would help me”, “I will use doping if it will help me with no negative health consequences”, “Not sure”, “I do not intend to engage in doping in the future”), (iv) their personal opinion about the main problem of doping (“Doping is a health hazard”, “Doping is against fair play”, “Not sure that doping should be forbidden”, “Doping should be allowed”), and (v) their personal opinion about doping penalties (“Lifelong suspension”, “First time milder punishment, then life-long suspension”, “Suspension for a couple of seasons”, “Financial punishment”, “Doping should be allowed”). As previous studies identified specific associations between the use of dietary supplements and doping behavior [32,34], athletes were asked about the use of dietary supplements (responses included “Yes, I consume supplements regularly”, “Yes, from time to time”, and “No, I don’t use dietary supplements”).

The coaching strategy and training methodology (CS&TM) was assessed through questions that were divided into three topics. Questions regarding the training methodology included swimmers’ perceptions about (i) general characteristics of their training, (ii) attention paid to mastering the swimming technique during training, (iii) training volume they were exposed to (i.e., swim distance) [35], and (iv) their characteristic training intensity. The coaching strategy was assessed by using the following “Yes–No” statements: (i) “Coach frequently explains the training aims”, (ii) “Coach overviews and discusses the quality of (my) execution of specific tasks”, (iii) “Coach is very strict and rigid”, (iv) “Discipline is an important part of our training regimen”, (v) “Coach pushes me very hard”, and (vi) “Sometimes, I don’t know what the Coach wants me to do in training”. General characteristics of the training were assessed by using the following three statements on a binomial (Yes–No) scale: “Swimming technique is an important part of my training”, “Training is monotonous and lacks diversity”, and “Training is mostly oriented toward volume (swim distance)”. The attention paid to technique during training was asked by one question (“The swimming technique is practiced …”), and swimmers had to choose one of three responses (“… in less than 10% of training”, “… in 10–30% of training”, or “… in more than one-third of training”). Swimmers self-reported their training volume on one question (“My average training volume is…”) and had to choose one of five possible answers (“… approximately 20–30 km per week”, “… 30–40 km per week”, “… 40–50 km per week”, “… 50–60 per week”, or “… >60 km per week”). The question on self-estimated “Training intensity” included six possible responses (“Training is high in intensity when I have to swim >6 km per session”, “… >2 km in one sequence”, “… repeated sets of maximal intensity, regardless of distance”, “… different relays while being highly focused on stroke technique, speed and force”, “… some specific sets which I have never/rarely performed before”, and “Intensity is high but with no specific reason”).

Knowledge about sports nutrition (KSN) was assessed through 10 questions (statements), each in a “true or false” format, and one point was scored for each correct answer. The ending results ranged from 0 to 10. The questions were as follows. (1) “The negative side effects of excessive sweating are best cured by drinking pure water.” (2) “After a competition day is over, it is better to not eat for 4 hours after a competition.” (3) “Dark yellow urine is a sign of proper hydration of the body.” (4) “For the first meal after a competition, chicken breast (white meat) and eggs are a better choice than pasta.” (5) “Dried fruit is an excellent source of carbohydrates.” (6) “Protein supplementation requires an increased intake of water.” (7) “Fresh fruit and vegetables are the best source of high-quality proteins.” (8) “Egg yolks and poultry are valuable sources of vitamins B and C.” (9) “Carbohydrate-rich meals should be avoided before competitions because they encourage urination and therefore dehydration.” (10) “A decrease in body weight as a result of a single training day indicates dehydration.” Items 1, 3, and 10 examined the knowledge of hydration/dehydration; questions 2, 4, and 6 targeted the knowledge of nutrition strategies aimed at recovery; and questions 5, 7, 8, and 9 were general questions about nutrition.

Knowledge about doping (KD) was assessed through ten questions. The same evaluation system previously explained for KSN was used. The accuracy of the results (answers) was based on WADA standards. The KD questions were as follows. (1) “Diuretics are considered doping because of their influence on body weight reduction.” (2) “Doping control officers should notify athletes of their testing intentions a few hours prior to any testing.” (3) “If an athlete has an out-of-competition doping test, four weeks should elapse before their next doping test.” (4) “If a doping control officer does not provide valid proof of identity, an athlete can refuse to participate in the testing.” (5) “A masking agent” is someone who helps an athlete hide their use of doping and is therefore equally responsible for doping offenses.” (6) “The use of amphetamines has been related to several cases of death in sport due to cardiovascular failure.” (7) “The use of amphetamines by women is related to male-like changes in the body appearance.” (8)” Synthetic testosterone (i.e., steroids) increases the number of erythrocytes and is therefore common in endurance sports and not prevalent in strength/power sports.” (9) “The use of synthetic testosterone (i.e., steroids) inhibits the production of natural (endogenous) testosterone.” (10) “When an athlete reports undergoing official medical treatment, he/she cannot be tested for doping.” Knowledge about doping side effects was asked by items 5, 6, 7, 8, and 9, whereas items 1, 2, 3, 4, and 10 targeted knowledge about the antidoping regulations.

### 2.4. Statistics

Normality of the distribution was checked by Kolmogorov Smirnov test. Descriptive statistics included means and standard deviations (for parametric variables), and counts and percentages (for nonparametric variables). Gender differences in studied variables were established by t-test for independent sample (for parametric variables), Chi-square test (χ^2^), and Mann–Whitney test (MW).

To establish relationships between the predictors and the criterion (PDB), a multinomial regressions were calculated, which included three possible responses in regard to the criterion: (1) negative PDB (those who responded, “I do not intend to engage in doping in the future.”), (2) neutral PDB (“not sure”), and (3) positive PDB (“I would engage in doping if it would help me.”). The negative PDB was set as the reference value. Significant associations between sociodemographic factors and personal opinions about the presence of doping in sports and PDB have been frequently reported in previous studies, and therefore, multinomial regressions were adjusted for age in order to examine the effect of age as a possible confounding factor [20,36]. The multinomial regressions were calculated separately for boys and girls. For all analyses, Statistica 13.0 (Dell, Tulsa, OK, USA) was used and *p* < 0.05 was applied.

## 3. Results

Boys were approximately 10 months older than girls (*t*-test: 5.68, *p* < 0.01) and were more experienced in swimming (*t*-test: 2.69, *p* < 0.01) (Table 1).

Boys used dietary supplements more often than girls (KW: 2.15, *p* < 0.05). Specifically, 18% of boys regularly consume dietary supplements and an additional 47% declared irregular usage. There was no significant difference between boys and girls in PDB, with altogether 70% of swimmers who declared negative-opinion about their future PDB. Further, majority of youth swimmers observed doping as mainly a problem of fair play (64%), while only one-third judged doping as mainly a health-threatening behavior (Table 2).

Girls more often than boys judged their training as being “monotonous and lacks diversity” than boys (χ^2^: 14.59, *p* < 0.01). Also, girls more often evidenced “discipline as important part of their training regime” (χ^2^: 6.91, *p* < 0.01), and stated that “coach pushes them very hard” (χ^2^: 4.97, *p* < 0.05). Finally, girls more frequently observed as highly intensive those training sessions when they have to swim: (i) “sets they have never performed before” (χ^2^: 22.32, *p* < 0.01), and (ii) “different relays while being highly focused on stroke technique, speed, and force” (χ^2^: 8.11, *p* < 0.05) (Table 3).

The KSN was the only factor related to PDB in boys, with lower doping susceptibility (negative-PDB) in boys who achieved better results on KSN (OR: 0.77, 95%CI: 0.60–0.95 for positive-PDB) (Table 4).

Those girls who achieved better competitive results in Olympic-pools (OR: 0.39, 95%CI: 0.24–0.63), and non-Olympic pools (OR: 0.62, 95%CI: 0.39–0.97) were less likely to declare neutral-PDB. The onset of intensive training was correlated with positive- (OR: 0.90, 95%CI: 0.83–0.99) and neutral-PDB (OR: 0.88, 95%CI: 0.81–0.96), and those girls who started with intensive training later in life were less likely to declare doping susceptibility (Table 5).

## 4. Discussion

There were a few important findings in this investigation. First, the associations between the CS&TM variables and potential doping behavior in junior swimmers were nonsignificant. Second, the results confirmed a lower susceptibility to doping in girls who were more successful (i.e., those who achieved better competitive results). Third, girls who initiated an intensive training regimen at a younger age (started earlier with training eight times a week) were more prone to PDB. Finally, boys who had better knowledge about nutrition and dietary supplementation were less prone to doping. Collectively, we may accept our initial study hypothesis (i.e., significant association between studied predictors and PDB). Because this is the first study that identified factors associated with PDB exclusively in high-level youth swimmers, we will first overview the prevalence of PDB in our participants.

### 4.1. Potential Doping Behavior in Youth Swimmers

Our results indicated that 71% of swimmers had a negative doping tendency. In previous studies where authors used the same measurement tools, a lower tendency toward PDB was found in sailing athletes (82%) and tennis players (75%), and a somewhat higher tendency was found in team sport athletes and synchronized swimmers (approximately 62–63% with negative PDB) [1,20,37]. Conversely, a much higher likelihood of doping has been reported for weightlifters, kickboxers, and rugby players (30%, 45%, and 51.4% negative PDB, respectively) [1,2,32]. The results of the doping tendency of youth swimmers in the present study are comparable to previous studies, where authors reported similar values for adult swimmers (>18 years): approximately 80% of swimmers had a negative tendency toward doping [23,29]. Altogether, these results place swimming among the sports with a high risk of doping behavior, which is directly supported by 2016 WADA reports of positive analytical findings [3]. There are several possible explanations for such findings.

The first explanation on relatively high doping susceptibility is related to characteristics of the swimming sport. The performance goal is to swim a given distance in the shortest time, and this is determined by the interaction of morphological, physiological, psychological, and technical factors that are based on individual genetic endowments [26,29]. Apart from certain genetically determined factors, the competitive achievement in swimming also depends on the training process, which influences achievement through training volume, training intensity, and mastering a specific swimming technique [38]. If we take into consideration that doping in sports is mostly used to enhance athletes’ physiological capacities (i.e., to overcome physiological stress induced by training volume and intensity and to boost the mechanism of supercompensation) [38], the high prevalence of PDB might be observed as logical, although disturbing.

Authors of the study are the opinion that the second explanation on alarming figures of PDB in youth swimmers is related to connection which exists between doping susceptibility and perception of swimming as doping contaminated sports. More specifically, previous studies found a higher doping susceptibility in athletes who perceive their sport as being doping contaminated [20,32,37,39], and these findings have been explained by the socio-psychological theory of self-categorization [40]. Accordingly, it is more likely that athletes will engage in doping in the future if he/she perceives their sport as being doping contaminated [1,41]. Consequently, the fact that a great percentage of swimmers think that their sport is doping contaminated (proximately 30% of youth swimmers believe that doping is common in their sport) may be a strong indicator of a highly corrupted social environment. If we take into consideration that people adopt the norms, beliefs, and behaviors of “their group” [42], this is another factor that could have boosted the prevalence of PDB in youth swimmers studied here.

### 4.2. Correlates of Potential Doping Behavior

The correlation between CS&TM variables and PDB in youth swimmers was not significant. Conversely, this was not the case in a previous study examining adult swimmers, where the authors reported several significant relationships between the CS&TM variables and PDB [23]. In short, adult swimmers (>18 years) from the same country (e.g. Slovenia) who perceived training as “monotonous and more oriented toward volume (swam distance)” were more susceptible to PDB, as were swimmers who perceived their coaches to be “indifferent and nonchalant in regard to athletes’ performance” [23]. The difference between our study and the previous study conducted with adult swimmers is not surprising if we consider that an awareness of training methodology and coaching strategy comes with experience. It is reasonable to assume that adult athletes have a greater number of coaches during their sporting career, and consequentially develop a more critical approach toward CS&TM. In contrast, youth swimmers probably do not tend to re-examine their coaching strategy and methodology, because of a high respect toward their coaches’ authority and a lack of experience, which together does not allow them to develop a critical opinion toward CS&TM.

Lower competitive achievement was evidenced as a risk factor for PDB in girls. In general, the association between sports results and PDB has been commonly studied, but the results are not fully consistent [2,19,20,28,31,32]. For instance, in a study conducted with sailing athletes [31], rugby players [32], and male team sport athletes [20], the authors reported a lower susceptibility to doping in more successful athletes. Additionally, results from studies done on male and female kickboxers [2] and female team sport athletes [20] found no correlations between competitive results and PDB. Because of such relatively inconsistent findings, it is hard to speculate the true logic underlying the “gender-specific”correlation between the achieved results and PDB in our study (i.e., a significant correlation in females, exclusively). However, discontent with the achieved results can be observed as the logical cause of potential doping behavior [28]. Athletes who are dissatisfied with their results will be probably more prone to doping on the grounds that they are eager to enhance their performance, which is particularly possible in highly objective sports similar to swimming.

Male swimmers who achieve higher scores regarding KSN are less likely to engage in doping in the future. Although this was not the first study to examine the association between KSN and PDB, the potential association between knowledge about nutrition and doping is a poorly explored topic, and previous reports examined this issue only in adult athletes [20,39]. Briefly, a study conducted with tennis players determined that there was a negative correlation between KSN and PDB in both males and females, whereas a more recent study examining team sports athletes found a negative correlation (i.e., a lower doping likelihood in those who had better knowledge) only for female athletes [20,39]. Our results suggest that a better knowledge of nutrition could have a certain preventive effect against PDB even in youth age. The possible mechanism is explained in the following text.

It is generally accepted that nutritional knowledge is imperative for athletes [43]. Although quality food choices do not compensate for a lack of training or inferior physical abilities, they will help athletes to make the most of their potential [44]. The amount and timing of food intake and types of food eaten will help athletes withstand consistent, intensive training, and competition without experiencing chronic fatigue, injury, or illness [45]. Therefore, the positive effect of proper nutrition on athletes’ physical and even psychological capacities may help in keeping them away from potential doping use [39]. On the other hand, the lack of a correlation between KD and PDB is potentially related to the small variance in KD and the consequent influence of the “truncated variance” on the statistical significance of the association between KD (predictor) and PDB (criterion), which could potentially be the consequence of difficulty of KD questions for young athletes in comparison to adults [46].

One of the most important findings of our study is the association between the onset of an intensive training regimen and PDB. Specifically, female swimmers who began an intensive training regimen at a younger age were more prone to PDB. The authors of this study are of the opinion that such a relationship might be initiated by a psychological phenomenon known as “burnout syndrome” [47,48,49]. Briefly, in an athletic environment, burnout syndrome can be defined as a syndrome comprised of three symptoms: (i) physical and emotional exhaustion, (ii) a reduced sense of accomplishment, and (iii) sport devaluation [50]. The prevalence of burnout has become a serious issue, especially among young adolescent athletes [51,52,53]. With regard to fact that the onset of intensive training was found to be a significant correlate of PDB only in females, it is interesting that previous studies identified higher levels of burnout among females [54].

Specifically, young athletes who are pressured to accomplish great results are required to take part in intensive training regimens, compete regularly, and pursue their schooling, all while dealing with the biological changes that occur during adolescence [55,56]. These demands can be hard to cope with, and in some cases can lead to psychological and physical fatigue, anxiety, stress, and a decline in feelings of accomplishment, which usually leads to decreased athletic performance [50,57]. This altogether can result in frustration, and not surprisingly, may even result in a positive tendency toward doping behavior. Such an interaction between dissatisfaction and doping tendency has already been noted, even in samples of noncompetitive athletes [58].

We should also take in consideration that the inaccurate or improper selection of athletes could be the true cause of the established association between early onset of intensive training and PDB. Sometimes, because of an improper educational background or in the rush for results, coaches misinterpret early bloomers (biological accelerants) as athletes with superior talent (physical abilities). Selecting athletes whose potential is determined by their accelerated growth and development, and not their true talent, is one of the worst mistakes that could be made in youth sport. High and unrealistic expectations placed against youth athletes, in addition to results achieved while they were eventually superior in comparison to their biologically inferior peers, makes it hard to cope with failure, after their peers catch up in growth and development.

### 4.3. Limitations and Strengths

This study is based on self-reports; therefore, athletes may tend to give socially desirable answers, but we believe that the strict anonymity and study design decreased this possibility. More precisely, participants were not asked about any information which could be directly or indirectly connected to single person (i.e., they were not asked for specific date of birth, competitive results achieved were questioned using ordinal scale (please see Variables for details)). Moreover, testing was not organized individually, but athletes were tested in groups of at least five participants. Finally, questionnaires were opened one day after testing by investigators who did not participate in testing, and this was all clearly explained to athletes examined. However, in future researches about the problems where social-desirability bias may appear, repeated testing should be considered as a possible solution. Furthermore, PDB should not necessarily be considered as an indefinite tendency toward doping, so it is almost certain that the majority of athletes who declared positive PDB will never engage in doping in their sporting career. However, antidoping campaigns should target vulnerable athletes, and those who declare a tendency toward doping are almost certainly subjects of interest in that manner. Finally, although we intended to study only variables that could be objectively evaluated by a quantitative design, based on our results (i.e., a correlation between the early onset of intensive training and PDB), we are aware that the study lacks a qualitative approach, which would allow a more accurate identification of the relationships between the studied variables.

This is a rare study in that examined youth athletes involved in one specific sport, and observed their tendency toward doping and the factors associated with such tendencies. Also, our investigation was based on testing and methodology that has been previously applied in various samples of athletes; therefore, the results are objectively comparable to those previously reported. Finally, the literature includes some specific correlates of PDB that were not previously examined in youth athletes (i.e., coaching strategy and training methodology, knowledge on sports nutrition and doping). Thus, we believe that our results, although not the final word on the problem, will contribute to the knowledge in this field, and initiate future research.

## 5. Conclusions

The present study evidenced an alarming doping susceptibility in youth swimmers. Moreover, it is important to note that the tendency toward doping in youth swimmers was similar to that of adult swimmers. These results altogether highlight the necessity of an urgent intervention and the development of systematic antidoping educational programs in youth swimming. In doing so, special attention should be paid on (i) female youth swimmers who begin intensive training at an earlier age and (ii) girls who do not achieve high competitive results in swimming. Meanwhile, there are certain evidences that proper knowledge on nutrition should be observed as protective factor against PDB in boys.

Our results suggest that the early onset of intensive training and maturation status are specifically connected to PDB in youth swimmers, which could be correlated with burnout syndrome, and improper selection of the potentially talented swimmers. Although this is one of the first investigations that evidenced such association, future studies in youth athletes should pay particular attention to this problem and investigate the associations between the maturation status (i.e., early vs. late maturation), onset of intensive training, sport selection, and doping behavior in youth athletes.

## Figures and Tables

**Table 1 ijerph-16-04851-t001:** Descriptive statistics for parametric variables with differences between genders.

	Boys		Girls		*t*-test	
	Mean	SD	Mean	SD	*t*-value	*p*
Age (years)	15.28	1.13	14.41	1.23	5.68	0.01
Experience in swimming (years)	8.18	2.51	7.31	2.51	2.69	0.01
Start 2x per day (age)	11.91	3.55	11.17	2.99	1.78	0.08
Start 8x per week (age)	12.06	4.73	11.16	4.31	1.56	0.12
KD (score)	1.71	1.49	1.39	1.38	1.69	0.1
KSN (score)	4.23	2.37	4.27	2.06	0.11	0.91

LEGEND: Start 2x per day—the age at which athletes started to participate in two training sessions per day. Start 8x per week—the age at which athletes started to participate in 8 and more training sessions per week. KD—knowledge on doping (scale range 0–10). KSN—knowledge on sport nutrition and dietary supplementation (scale range 0–10).

**Table 2 ijerph-16-04851-t002:** Descriptive statistics for observed sport-specific and doping-related categorical and ordinal variables with differences between genders (Mann–Whitney test (MW) or Chi-square test: χ^2^).

Variables	Boys		Girls		MW/χ^2^ (p)
	f	%	f	%	
Competitive result Olympic pools					0.75 (0.44)
Regional level	16	14.41	19	14.50	
National finals	35	31.53	38	29.01	
National medal	46	41.44	57	43.51	
International finals	5	4.50	3	2.29	
International medal	4	3.60	3	2.29	
Missing	5	4.50	11	8.40	
Competitive result non-Olympic pools					0.08 (0.93)
Regional level	19	17.12	21	16.03	
National finals	41	36.94	41	31.30	
National medal	44	39.64	57	43.51	
International finals	1	0.90	2	1.53	
International medal	2	1.80	1	0.76	
Missing	4	3.60	9	6.87	
Doping testing					1.31 (0.18)
Never tested on doping	1	0.90	122	93.13	
once or twice	106	95.50	9	6.87	
3–5 times	2	1.80	0	0.00	
>5 times	2	1.80	0	0.00	
Missing	0	0.00	0	0.00	
Doping in swimming					1.23 (0.22)
I don’t think doping is used	6	5.41	2	1.53	
I’m not sure	11	9.91	16	12.21	
Used, but rarely	55	49.55	82	62.60	
Doping is often	39	35.14	31	23.66	
Missing	0	0.00	0	0.00	
Penalties for doping offenders					0.19 (0.85)
First time, lifelong suspension	20	18.02	30	22.90	
First time milder punishment, then lifelong suspension	49	44.14	49	37.40	
Suspension for several seasons	40	36.04	49	37.40	
Financial penalties	2	1.80	3	2.29	
Doping should be allowed	0	0.00	0	0.00	
Missing	0	0.00	0	0.00	
Potential doping behavior					0.76 (0.44)
I will use doping if it will help me	5	4.50	3	2.29	
I will use it if it will help me with no negative health cons	13	11.71	7	5.34	
Not sure	16	14.41	27	20.61	
I will not use doping	77	69.37	94	71.76	
Missing	0	0.00	0	0.00	
Potential doping behavior (tendency)					0.77 (0.44)
Negative	77	69.37	94	71.76	
Neutral	16	14.41	27	20.61	
Positive	18	16.22	10	7.63	
Missing	0	0.00	0	0.00	
Main problem of doping (χ^2^)					0.52 (0.77)
Doping is health hazard	37	33.33	48	36.64	
Doping is against fair play	73	65.77	81	61.83	
I’m not sure that doping should be banned	1	0.90	2	1.53	
Doping should be allowed	0	0.00	0	0.00	
Missing	0	0.00	0	0.00	
Usage of the dietary supplements					2.15 (0.03)
Yes, regularly	20	18.02	8	6.11	
Yes, from time to time	52	46.85	66	50.38	
No	39	35.14	57	43.51	
Missing	0	0.00	0	0.00	

**Table 3 ijerph-16-04851-t003:** Descriptive statistics for variables of coaching strategy and training methodology with differences between genders (Mann–Whitney test (MW) or Chi square test: χ^2^).

Variables	Boys				Girls				MW/χ^2^ (p)
	NO		YES		NO		YES		
	F	%	F	%	F	%	F	%	
General Opinion About Training									
Technique is an important part of my training (χ^2^)	43	39	68	61	46	35	85	65	0.33 (0.56)
Training is monotonous and lacks diversity (χ^2^)	68	61	43	39	48	37	83	63	14.59 (0.01)
Training is mostly oriented toward volume (χ^2^)	55	50	56	50	51	39	80	61	2.75 (0.09)
Intensity High When I Have to Swim …									
… >6 km per session (χ^2^)	35	32	76	68	40	31	91	69	0.03 (0.87)
… >2 km in one sequence (χ^2^)	51	46	60	54	67	51	64	49	0.65 (0.42)
… repeated sets of maximal intensity	61	55	50	45	71	54	60	46	0.01 (0.99)
… different relays while being highly focused on stroke technique, -speed, and –force (χ^2^)	44	40	67	60	76	58	55	42	8.11 (0.04)
… sets I have never/rarely performed before	24	22	87	78	67	51	64	49	22.32 (0.01)
Intensity is high but with no specific reason (χ^2^)	19	17	92	83	21	16	110	84	0.05 (0.83)
Coaching									
Coach frequently explains the training aims (χ^2^)	81	73	30	27	94	72	37	28	0.04 (0.83)
Coach overviews and discuss the quality of (my) execution of specific tasks (χ^2^)	78	70	33	30	88	67	43	33	0.26 (0.60)
Coach is very strict and rigid (χ^2^)	65	59	46	41	78	60	53	40	0.02 (0.87)
Discipline is an important part of our training regime (χ^2^)	80	72	31	28	73	56	58	44	6.91 (0.01)
Coach pushes me very hard (χ^2^)	82	74	29	26	79	60	52	40	4.97 (0.03)
Sometimes, I don’t know what does the Coach wants me to do in training (χ^2^)	34	31	77	69	45	34	86	66	0.37 (0.54)
Training Volume									0.09 (0.93)
Average volume is ~20–30 km per week			18	16			21	16	
Average volume is ~30–40 km per week			32	29			34	26	
Average volume is ~40–50 km per week			34	31			47	36	
Average volume is ~50–60 km per week			16	14			15	11	
Average volume is >60 km per week			9	8			12	9	
Missing (don’t know)			2	2			2	2	
Technique (Approximation)									0.19 (0.85)
Technique is practiced in less than 10% of training			20	18			23	18	
Technique is practiced in 10–30% of training			41	37			49	37	
Technique is practiced in more than one-third of training			40	36			49	37	
Missing (don’t know)			10	9			12	9	

**Table 4 ijerph-16-04851-t004:** Correlates of potential doping behavior (PDB) in boys with negative PDB being set as referent value.

Variables	Positive-PDB	Neutral-PDB
	OR (95%CI)	OR (95%CI)
	0.72 (0.43–1.21)	1.04 (0.60–1.79)
Competitive result non-Olympic pools ^cont^	0.6 (0.32–1.10)	1.32 (0.70–2.49)
Trust on officials about doping *	0.95 (0.55–1.63)	1.23 (0.81–1.86)
Doping testing ^cont^	1.06 (0.15–7.56)	0.85 (0.17–4.08)
Doping in swimming ^cont^	2.73 (1.06–7.03)	1.57 (0.87–2.85)
Penalties for doping offenders ^cont^	1.77 (0.71–4.43)	1.34 (0.68–2.66)
Main problem of doping		
Doping is health hazard	0.68 (0.42–1.12)	1.08 (0.67–1.77)
Doping is against fair play	0.59 (0.42–1.15)	1 (0.80–1.41)
I’m not sure that doping should be banned	REF	REF
Usage of the dietary supplements		
Yes, regularly	0.62 (0.24–1.63)	1.31 (0.63–2.73)
Yes, from time to time	1.02 (0.50–2.78)	1.04 (0.20–7.81)
No	REF	REF
Experience ^cont^	0.91 (0.74–1.13)	1.09 (0.85–1.41)
Start 2x per day ^cont^	0.93 (0.82–1.05)	1.09 (0.88–1.35)
Start 8x per week ^cont^	0.99 (0.91–1.10)	1.3 (0.91–1.86)
KD ^cont^	0.80 (0.55–1.17)	1.15 (0.81–1.64)
KSN ^cont^	0.77 (0.60–0.95)	1.25 (0.97–1.60)
Technique is an important part of my training *	0.25 (0.05–1.25)	1.72 (0.41–7.23)
Training is monotonous and lacks diversity *	2.02 (0.48–8.43)	1.66 (0.54–5.13)
Training is mostly oriented toward volume *	0.48 (0.12–1.89)	0.29 (0.10–1.20)
Intensity is high when I have to swim >6 km per session *	1.22 (0.31–4.80)	1.69 (0.59–4.84)
Intensity is high when I have to swim >2 km in one sequence *	NC	NC
Intensity is high when I have to swim repeated sets of maximal intensity *	0.33 (0.03–3.58)	0.27 (0.05–1.35)
Intensity is high when I have to swim different relays while being highly focused on stroke technique, speed, and force *	1.13 (0.06–19.74)	0.45 (0.04–5.29)
Intensity is high when I have to swim sets I have never/rarely performed before *	0.87 (0.18–4.00)	0.58 (0.17–1.88)
Intensity is high but with no specific reason *		
Coach frequently explains the training aims *	0.85 (0.19–3.71)	1.45 (0.45–4.72)
Coach overviews and discuss the quality of (my) execution of specific tasks *	NC	NC
Coach is very strict and rigid *	0.28 (0.07–1.26)	0.71 (0.21–2.41)
Discipline is an important part of our training regime *	3.00 (0.61–14.87)	2 (0.53–7.59)
Coach pushes me very hard *	0.39 (0.99–1.56)	0.93 (0.31–2.74)
Sometimes, I don’t know what does the Coach wants me to do in training *	NC	NC
Training volume ^cont^	5.67 (0.56–57.23)	0.45 (0.03–5.29)
Technique ^cont^	2.27 (0.45–11.59)	1.31 (0.34–5.09)

LEGEND: * denotes variables where response “No” was set as referent value in regression calculation, ^cont^ denotes variables observed as continuous for the purpose of regression calculation, REF—referent value, NC—not calculated because of the matrix singularity.

**Table 5 ijerph-16-04851-t005:** Correlates of potential doping behavior (PDB) in girls with negative PDB being set as referent value.

Variables	Positive-PDB	Neutral-PDB
	OR (95%CI)	OR (95%CI)
Competitive result Olympic pools ^cont^	0.62 (0.31–1.22)	0.39 (0.24–0.63)
Competitive result non-Olympic pools ^cont^	0.63 (0.33–1.21)	0.62 (0.39–0.97)
Trust on officials about doping *	0.96 (0.48–1.94)	1.01 (0.54–1.93)
Doping testing ^cont^	NC	0.32 (0.05–1.81)
Doping in swimming ^cont^	0.42 (0.13–1.41)	0.58 (0.20–1.74)
Penalties for doping offenders ^cont^	2.78 (1.07–7.21)	1.76 (0.76–4.09)
Main problem of doping		
Doping is health hazard	1.88 (0.45–7.82)	1.83 (0.51–6.56)
Doping is against fair play	0.90 (0.42–6.21)	0.99 (0.21–7.29)
I’m not sure that doping should be banned	REF	REF
Usage of the dietary supplements		
Yes, regularly	7.4 (0.39–137.84)	0.73 (0.30–1.77)
Yes, from time to time	7.9 (0.95–67.16)	1.01 (0.10–1.90)
No	REF	REF
Experience ^cont^	0.98 (0.74–1.31)	0.96 (0.79–1.17)
Start 2x per day ^cont^	0.92 (0.77–1.10)	0.97 (0.84–1.13)
Start 8x per week ^cont^	0.9 (0.83–0.99)	0.88 (0.81–0.96)
KD ^cont^	1.19 (0.75–1.89)	1.07 (0.78–1.46)
KSN ^cont^	1.02 (0.72–1.46)	0.91 (0.73–1.13)
Technique is an important part of my training *	0.42 (0.09–2.03)	0.33 (0.08–1.30)
Training is monotonous and lacks diversity *	NC	NC
Training is mostly oriented toward volume *	1.22 (0.25–6.08)	1.58 (0.37–6.69)
Intensity is high when I have to swim >6 km per session *	NC	NC
Intensity is high when I have to swim >2 km in one sequence *	1.13 (0.26–5.01)	1.17 (0.31–4.46)
Intensity is high when I have to swim repeated sets of maximal intensity *	0.98 (0.20–4.79)	0.85 (0.20–3.52)
Intensity is high when I have to swim different relays while being highly focused on stroke technique, speed, and force *	3.2 (0.57–17.97)	6.44 (1.29–32.06)
Intensity is high when I have to swim sets I have never/rarely performed before *	0.33 (0.07–1.46)	0.35 (0.07–1.13)
Intensity is high but with no specific reason *	0.15 (0.01–1.92)	0.13 (0.02–1.08)
Coach frequently explains the training aims *	NC	NC
Coach overviews and discuss the quality of (my) execution of specific tasks *	NC	NC
Coach is very strict and rigid *	1.45 (0.34–6.25)	1 (0.27–3.68)
Discipline is an important part of our training regime *	0.35 (0.08–1.58)	0.77 (0.21–2.85)
Coach pushes me very hard *	NC	0.29 (0.03–3.16)
Sometimes, I don’t know what does the Coach wants me to do in training *	NC	NC
Training volume ^cont^	0.66 (0.13–3.39)	1.09 (0.26–4.53)
Technique ^cont^	0.4 (0.07–2.26)	0.37 (0.08–1.63)

LEGEND: * denotes variables where response “No” was set as referent value in regression calculation, ^cont^ denotes variables observed as continuous for the purpose of regression calculation, REF—referent value, NC—not calculated because of the matrix singularity.

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
