# Peer review of "Toward Prevention of Doping in Youth Sport: Cross-Sectional Analysis of Correlates of Doping Tendency in Swimming"

_ijerph, 2019, doi:10.3390/ijerph16234851_

Round 1
Reviewer 1 Report
Broad Comments
The topic is of scientific interest and the outcomes may find potential practical application in the fight against doping since they highlight the necessity of systematic anti-doping educational programs in youth athletes.
Specific Comments
Were there any prior statistical power analyses for the sample of the study? The ref. no 3 should be updated (you use the report of 2016 while the relative report of 2017 is available for many months now). The latest report of 2017 does not show an overall growing trend in the percentage of adverse analytical findings (it is below 1.5%). Nevertheless, consider using also other WADA estimates stating that the true percentage of doping use is over 10%. In line 39, you mention that the number of adverse analytical findings is increasing but keep in mind that also the total number of annually analyzed samples is increasing. Please consider mentioning just the percentages. You mention the past anti-doping testing figures report of 2016 in lines 38,39, 90 and 292 of your manuscript. Consider using the supplementary table instead of table 2 in the text since the supplementary table is both more relevant with the title of your paper and the overall scope of the study. The 10 questions about the KD might be appropriate for adult athletes but it seems to be of increased difficulty for adolescents and this might be another reason for not finding any correlation between KD and PDB. I suggest that you should discuss this remark in the limitations section of your paper. Lines 360-362, consider make an hypothesis and not stating “….ACTUALLY initiated by a psychological phenomenon known as “burnout syndrome”. Use “might” instead of “actually” since the burnout syndrome was not really diagnosed but just speculated. The English language is OK but a revision by a native English speaker would improve the overall quality of the paper.Author Response
Broad Comments
The topic is of scientific interest and the outcomes may find potential practical application in the fight against doping since they highlight the necessity of systematic anti-doping educational programs in youth athletes.
RESPONSE: Thank you for your support and for recognizing the potential of our manuscript. Please see bellow how we responded to your suggestions and where to find specific amendments. Staying at your disposal.
Specific Comments
Were there any prior statistical power analyses for the sample of the study?
RESPONSE: Thank you for your question, this was not clearly stated in text however we did not calculate statistical power because the sample of swimmers in our study represented the whole population of competitive youth swimmers in Slovenia. Text is amended and now reads: “The sample included all youth age swimmers (<18 years) who participated in the National Championship for the year of 2017. Therefore, the total population of competitive youth age swimmers from the country was included/represented.” (please see Materials and Methods - Participants – 1st paragraph)
The ref. no 3 should be updated (you use the report of 2016 while the relative report of 2017 is available for many months now). The latest report of 2017 does not show an overall growing trend in the percentage of adverse analytical findings (it is below 1.5%). Nevertheless, consider using also other WADA estimates stating that the true percentage of doping use is over 10%. In line 39, you mention that the number of adverse analytical findings is increasing but keep in mind that also the total number of annually analyzed samples is increasing. Please consider mentioning just the percentages. You mention the past anti-doping testing figures report of 2016 in lines 38,39, 90 and 292 of your manuscript.
RESPONSE: Thank you for this suggestion, text is amended and now reads: “Although the World Anti-Doping Agency (WADA) has been exploring and developing the most efficient ways to improve doping statistics, data from WADA anti-doping rule violation reports don’t provide evidence of decrease (2015: 1.26%, 2016: 1.60% and 2017: 1.43%) while WADA approximates that a true number exceeds 10% [3,4]. (please see Introduction – 1st paragraph)
Consider using the supplementary table instead of table 2 in the text since the supplementary table is both more relevant with the title of your paper and the overall scope of the study.
RESPONSE: As you suggested, the Supplementary table is now included in the main manuscript file (now Table 2). Thank you.
The 10 questions about the KD might be appropriate for adult athletes but it seems to be of increased difficulty for adolescents and this might be another reason for not finding any correlation between KD and PDB. I suggest that you should discuss this remark in the limitations section of your paper.
RESPONSE: Thank you for this suggestion, text is amended and now reads: “On the other hand, the lack of a correlation between KD and PDB is potentially related to the small variance in KD and the consequent influence of the “truncated variance” on the statistical significance of the association between KD (predictor) and PDB (criterion), which could potentially be the consequence of increased difficulty of KD questions for young athletes in comparison to adults.” (please see Discussion - Correlates of potential doping behavior – 4th paragraph)
Lines 360-362, consider make an hypothesis and not stating “….ACTUALLY initiated by a psychological phenomenon known as “burnout syndrome”. Use “might” instead of “actually” since the burnout syndrome was not really diagnosed but just speculated. The English language is OK but a revision by a native English speaker would improve the overall quality of the paper.
RESPONSE: Thank you for this suggestion, text is amended and now reads: “One of the most important findings of our study is the association between the onset of an intensive training regimen and PDB. Specifically, female swimmers who began an intensive training regimen at a younger age were more prone to PDB. The authors of this study are of the opinion that such a relationship might be initiated by a psychological phenomenon known as “burnout syndrome.”(please see Discussion - Correlates of potential doping behavior – 5th paragraph)
Thank you once again!
Staying at your disposal for any further amendments.
Authors
Reviewer 2 Report
Doping is a scourge for sport, not only because of the consequences for the health of the athlete, but also because of the deterioration that it is produced in the public image of the sport and in the transmission of harmful and immoral ideals to society, mainly to youth. The authors argue that swimming is one of the most contaminated sports by doping. They indicate that there are no studies that reliable determine factors associated with doping behavior (and tolerance to doping practices) in young swimmers, being this the main objective of the MS. The authors also emphasize that youth is a critical phase of personal development, when life attitudes are established, so adolescence is an essential period to induce ethical ideals in sport, limiting the impact of doping.
In order to achieve their goals, the authors assess the potential doping behavior in swimmers under 18 and their association with a variety of factors, such as sociodemographic factors, coaching and training strategies, knowledge about nutrition and association with the uptake of nutritional supplements, onset of intensive training…
Because the objective of this research is to detect possible factors that promote the use of fraudulent techniques in young people, this research has scientific, medical and social importance, its publication is highly recommended.
The main concern is that the research is based on a questionnaire made to a total of 242 young swimmers, 131 girls and 114 boys. Perhaps the main problem with the study that it relies on a criterion (potential doping behavior) to evaluate various predictors. The young swimmers, based on an answer, are categorized as positive (response: ‘I would engage in doping if it would help me’), neutral (response: ‘not sure’) and negative (response: ‘“I do not intend to engage in doping in the future). One of the main problems of using questionnaires as a research tool is the appearance of bias, particularly in this case, where the response to the criterion is the basis of the study and the form of assessment of the predictors. Although the authors mentioned that in the last part of the discussion, it would be interesting to know the opinion of the authors about how to avoid social desirability bias in their results. Another question which is intimately linked to the aforementioned comment it the repeatability of the test. Do the authors think that it would have been interesting to repeat the test?. Do the authors think that the repetition of the test would help to decrease the influence of the social opinion in the answers of the young swimmers?.
I would recommend avoiding very categorical sentences, if there is no exact scientific basis. The full manuscript should be reviewed for it. Sometimes the authors try to explain their results, which is correct, but because they do not have exact data concerning the explanations for their data, they should emphasize that these present only possible hypotheses or possible justifications of results. For instance, sentence of lines 347-348. The authors wrote: ‘Our results suggest that a better knowledge of nutrition has a certain preventive effect against PDB even in youth age’. This statement is controversial and not fully demonstrated, particularly when in the following paragraph the authors indicate that the variations in knowledge about nutrition was low and quite similar between the participants in the study.
Author Response
Doping is a scourge for sport, not only because of the consequences for the health of the athlete, but also because of the deterioration that it is produced in the public image of the sport and in the transmission of harmful and immoral ideals to society, mainly to youth. The authors argue that swimming is one of the most contaminated sports by doping. They indicate that there are no studies that reliable determine factors associated with doping behavior (and tolerance to doping practices) in young swimmers, being this the main objective of the MS. The authors also emphasize that youth is a critical phase of personal development, when life attitudes are established, so adolescence is an essential period to induce ethical ideals in sport, limiting the impact of doping.
In order to achieve their goals, the authors assess the potential doping behavior in swimmers under 18 and their association with a variety of factors, such as sociodemographic factors, coaching and training strategies, knowledge about nutrition and association with the uptake of nutritional supplements, onset of intensive training…
Because the objective of this research is to detect possible factors that promote the use of fraudulent techniques in young people, this research has scientific, medical and social importance, its publication is highly recommended.
RESPONSE: Thank you for recognizing the potential of our paper and support. In the following text we specified how we responded to your suggestions. Staying at your disposal. Authors
The main concern is that the research is based on a questionnaire made to a total of 242 young swimmers, 131 girls and 114 boys. Perhaps the main problem with the study that it relies on a criterion (potential doping behavior) to evaluate various predictors. The young swimmers, based on an answer, are categorized as positive (response: ‘I would engage in doping if it would help me’), neutral (response: ‘not sure’) and negative (response: ‘“I do not intend to engage in doping in the future). One of the main problems of using questionnaires as a research tool is the appearance of bias, particularly in this case, where the response to the criterion is the basis of the study and the form of assessment of the predictors. Although the authors mentioned that in the last part of the discussion, it would be interesting to know the opinion of the authors about how to avoid social desirability bias in their results.
RESPONSE: Thank you for your suggestion, in this version of manuscript we payed particular attention to this problem. Text is amended and now reads: “More precisely, participants were not asked about any information which could be directly or indirectly connected to single person (i.e. they were not asked for specific date of birth, competitive results achieved were questioned using ordinal scale [please see Variables for details]). Moreover, testing was not organized individually, but athletes were tested in groups of at least five participants. Finally, questionnaires were opened one day after testing by investigators who didn’t participate in testing, and this was all clearly explained to athletes examined.” (please see Limitations and strengths – 1st paragraph).
Another question which is intimately linked to the aforementioned comment it the repeatability of the test. Do the authors think that it would have been interesting to repeat the test?. Do the authors think that the repetition of the test would help to decrease the influence of the social opinion in the answers of the young swimmers?.
RESPONSE: Indeed, authors are of the opinion that this method could be used for verifying if athletes were answering randomly or were they reading answers and questions, and answering accordingly, on the other hand, there are simpler solutions for this. However, it is hard to assume if this could be useful for prevention of socially desirable answers, and we may only assume there is a slight possibility that if athletes know they have to complete questionnaire again they will tend to answer honestly instead of choosing more socially desirable answers, so they can easily recall previous answers, and respond the same way again. However, this possibility is briefly discussed in limitation subsection (please see end of the first paragraph of the Limitations subsection).
I would recommend avoiding very categorical sentences, if there is no exact scientific basis. The full manuscript should be reviewed for it. Sometimes the authors try to explain their results, which is correct, but because they do not have exact data concerning the explanations for their data, they should emphasize that these present only possible hypotheses or possible justifications of results. For instance, sentence of lines 347-348. The authors wrote: ‘Our results suggest that a better knowledge of nutrition has a certain preventive effect against PDB even in youth age’. This statement is controversial and not fully demonstrated, particularly when in the following paragraph the authors indicate that the variations in knowledge about nutrition was low and quite similar between the participants in the study.
RESPONSE: Thank you for your comment, the manuscript is systematically checked for conclusive sentences and amended accordingly. For example, the sentence you mentioned before now reads: “Our results suggest that a better knowledge of nutrition could have a certain preventive effect against PDB even in youth age. The possible mechanism is explained in the following text. It is generally accepted that nutritional knowledge is imperative for athletes., etc.”(please see Discussion - Correlates of potential doping behavior – 3 rd, 4th paragraph). Also, changes are done in some other part of the text. For example: “There are several possible explanations for such findings” (please see end of 2nd paragraph of the Discussion). “the high prevalence of PDB might be observed as logical, although disturbing. “ (end of 3rd paragraph of the Discussion). “Authors of the study are the opinion that the second explanation on alarming figures of PDB in youth swimmers is related to connection which exists between doping susceptibility and perception of swimming as doping contaminated sports.” (beginning of the 4th paragraph of the Discussion). “In contrast, youth-age swimmers probably do not tend to re-examine their coaching strategy and methodology, because of a high respect toward their coaches’ authority and a lack of experience, which together does not allow them to develop a critical opinion toward CS&TM. “; “However, discontent with the achieved results can be observed as the logical cause of potential doping behavior”: “Athletes who are dissatisfied with their results will be probably more prone to doping on the grounds that they are eager to enhance their performance, which is particularly possible in highly objective sports similar to swimming”
Thank you once again!
Staying at your disposal for any further amendments.
Authors